# Contact Tracing Strategies for COVID-19 Prevention and Containment: A Scoping Review

Bolanle Adefowoke Ojokoh [1], Benjamin Aribisala [2], Oluwafemi A. Sarumi [3,4,*], Arome Junior Gabriel [5], Olatunji Omisore [6], Abiola Ezekiel Taiwo [7], Tobore Igbe [8], Uchechukwu Madukaku Chukwuocha [9], Tunde Yusuf [10], Abimbola Afolayan [1], Olusola Babalola [4], Tolulope Adebayo [4] and Olaitan Afolabi [4]

1   Department of Information Systems, Federal University of Technology, Akure PMB 704, Ondo State, Nigeria
2   Department of Computer Science, Lagos State University, Ojo PMB 0001, Lagos State, Nigeria
3   Department of Mathematics and Computer Science, University of Marburg, 35037 Marburg, Germany
4   Department of Computer Science, Federal University of Technology, Akure PMB 704, Ondo State, Nigeria
5   Department of Cyber Security, Federal University of Technology, Akure PMB 704, Ondo State, Nigeria
6   Shenzhen Institutes of Advanced Technology, Chinese Academy of Sciences, Shenzhen 518055, China
7   Faculty of Engineering, Mangosuthu University of Technology, Durban 4026, South Africa
8   Center for Diabetes Technology, Department of Psychiatry & Neurobehavioral Sciences, University of Virginia, Charlottesville, VA 22903, USA
9   Department of Public Health, Federal University of Technology, Owerri PMB 1526, Imo State, Nigeria
10  Department of Mathematical Sciences, Federal University of Technology, Akure PMB 704, Ondo State, Nigeria
*   Correspondence: oluwafemi.sarumi@uni-marburg.de

**Abstract:** Coronavirus Disease 2019 (COVID-19) spreads rapidly and is easily contracted by individuals who come near infected persons. With this nature and rapid spread of the contagion, different types of research have been conducted to investigate how non-pharmaceutical interventions can be employed to contain and prevent COVID-19. In this review, we analyzed the key elements of digital contact tracing strategies developed for the prevention and containment of the dreaded epidemic since its outbreak. We carried out a scoping review through relevant studies indexed in three databases, namely Google Scholar, PubMed, and ACM Digital Library. Using some carefully defined search terms, a total of 768 articles were identified. The review shows that 86.32% (n = 101) of the works focusing on contact tracing were published in 2020, suggesting there was an increased awareness that year, increased research efforts, and the fact that the pandemic was given a very high priority by most journals. We observed that many (47.86%, n = 56) of the studies were focused on design and implementation issues in the development of COVID-19 contact tracing systems. In addition, has been established that most of the studies were conducted in 41 countries and that contract tracing app development are characterized by some sensitive issues, including privacy-preserving and case-based referral characteristics.

**Keywords:** contact tracing; COVID-19; design considerations; data ethics; user privacy

## 1. Introduction

Coronavirus Disease 2019 (COVID-19), caused by the severe acute respiratory syndrome–coronavirus2 (SARS-CoV-2), is a major trending medical condition that has received global attention since its onset until now [1]. Early studies established that the virus responsible for the new COVID-19 illness spread mostly by being close or in contact with infected patients; meanwhile, a person that is already affected can carry the virus for several days without showing symptoms. Thus, the virus grew very fast to the extent of reaching a daily incidence rate of approximately 900,000 cases (April, 2021) and a total of 263 million cases globally (https://covid19.who.int/ (accessed on 1 April 2021). In addition, the mortality rate increased regularly up until the first half of 2020 and many countries had to introduce nationwide lockdowns to reduce its spread. Different studies have been

conducted towards ameliorating the global society back to having a healthy socioeconomic status; however, outputs from such studies are still daunting and it remains unclear how the daily average toll of the disease can be eliminated. Although the first incidence of this pandemic was recorded in Wuhan, China, back in November 2019, human-to-human transmission remains the most important factor in the virus's accelerated global expansion. Many countries in Europe, Asia, and the United Kingdom implemented several restrictions to prevent the spread of COVID-19 whereas other countries such as the United States and Sweden allowed a less-restricted movement during the early time of the pandemic breakout [2]. Numerous efforts have been put in place for tracking the virus in different parts of the world. Nevertheless, it is still practically hard to predict future situations and spread patterns of the epidemic as new variants such as the Beta, Delta, and Omicron also induced subsequent waves of the epidemic. For instance, researchers are yet to identify how the virus mutates, how long new variants can remain potent in patients who may continue to transmit the virus and infect others, and when it will finally subside. These and many other questions are yet unanswered.

Many research and development projects have recently been implemented to produce drugs for monitoring and inhibiting the spread of COVID-19. In addition, pharmaceutical companies in developed countries have produced vaccines that can aid a lasting solution to COVID-19. However, success in developmental cases is yet to be documented with effective clinical outcomes. Yet, analyzing existing COVID-19 epidemiological data and the provision of effective containment measures have remained prominent among non-pharmaceutical researchers across the globe [3]. For instance, a retrospective cohort analysis of COVID-19 epidemiology and transmission in 391 individuals as well as 1286 close contacts was reported in Shenzhen, China [4]. Although the results were limited by the intricacies of human movement and interactions, further research work by other authors [5] identified that effective data-driven methods, predictive modeling, and effective containment measures could help policymakers make better-informed decisions that may well be useful in various parts of the world. Thus, the development of an intelligent predictive model can be tailored toward a suitable and effective containment measure and preventive approaches which can enhance policymaking at the governmental, medical, social, economic, and other stakeholder levels for taking the measures required to address COVID-19 concerns [6].

COVID-19 manifested when people's knowledge about it was limited. Initial challenges of diagnosing, delays in vaccine production, and global ill-declaration made the virus spread globally. Thus, studies on contact tracing were motivated as a way of identifying infected cases. The victims are traced to isolate them for treatment and reduce incidence. Most developed nations have deployed contact tracers via electronic devices such as mobile phones and location-based technologies, with Bluetooth and Wi-Fi capabilities being the most often used. Mobile telephony communication and mobility data were adopted for contact tracing during the Ebola epidemic in Sierra Leone [7]. The method involved using anonymized mobile data to track places that infected cases had visited via their mobile phones [8]. Mobile phone data is effective in tracking the transmission of infectious diseases, predominantly in densely populated and well-networked areas. Specifically, because the effect of case seclusion is based on minimizing contact of unaffected persons with index persons while they are sick, gathering and evaluating data while being exchanged in diverse social contexts is particularly helpful in plotting intervention strategies [9]. One of the ways of doing this is to locate or trace the infected using their medical records or travel history. Such approaches are good but could be enhanced by contact tracing technology. An early study that quantified SARS-CoV-2 transmission claimed that the virus may be managed through contact tracing [1]. However, the COVID-19 epidemic has been described as the largest community health problem in the last century [9]. COVID-19, unlike prior infectious diseases, is difficult to manage using typical manual tracing methods. Previous research efforts have revealed that contact tracing systems developed on manual data curation, structured data from network providers, and national disease control centers



can be utilized for tracing and containment of COVID-19. According to research work by Kedia et al. [10] and Schmidtke [11], such information can be logically presented to the general public via a graphical user-friendly interface for awareness purposes. Thus, it is necessary to review the early studies on contact tracing systems to identify the knowledge gap and justify the need for a better approach with enhanced data safety, patient privacy, and platform usage flexibility. This will increase people's awareness and offer ways to further prevent the spread of the virus.

Today, mobile phones are designed with many multiple sensors that have powerful features [12]. Aside from having Bluetooth sensors, some smartphones are embedded with a digital compass, accelerometer, GPS sensors, humidity sensors, and a camera for health tracking sensors. In addition, the microphone feature can aid telehealth and assistant systems. These additional devices can provide multimodal data upon which computer algorithms can operate. Lately, machine learning methods have been demonstrated to have great potential for modeling the spread of the virus and predicting epidemics [13]. Intelligent mobile applications have recently been adopted to track and contain the spread of COVID-19. For instance, Walrave et al. [14] developed a mobile app that operates a health belief model for identifying when mobile users make contact with active carriers of SARS-CoV-2. Thus, the system could provide timely notice to owners and warn them when at risk of contracting the virus. However, according to Yasaka et al. [15], the usefulness of these apps is highly dependent on their adoption by the masses. Vedaei et al. [16] presented a potential Internet of Things (IoT) application for proximity monitoring during pandemics. A lightweight and low-cost IoT node, fog-based machine learning algorithms (for data analysis and diagnostics), and a smartphone application, makes up the solution. A related framework was developed for sustainable contact tracing and applied for extensive investigation of COVID-19 exposure [17]. As the virus continues to rampage in the different parts of the globe, many Pacific Asian and Antarctica nations such as China, New Zealand, Thailand, and South Korea returned to normality in early 2021. However, the situation is different in African countries where both incidence and prevalence are less monitored. As a result of the poor state of health care delivery systems, the incidence rates are predicted to worsen over time in Africa. Challenges in these regions are the overwhelming workload healthcare workers face during contact tracing (typically done manually), and routines involved in isolation and quarantine [18].

Developing automated contact tracing apps for monitoring COVID-19 was hindered by several sensitive issues such as mobile users' security, data privacy, and mental health issues. Zhang et al. [19] discussed the perceptions of Americans about privacy and surveillance all through this dreaded pandemic. In the Netherlands, as in many other countries, ideas for contact tracing apps serve as a way of assisting in the containment of COVID-19 from further spreading. As a result, concerns about the usage of tracing apps in [20] have been outlined concerning the choice between privacy and public health. Furthermore, scaling such developments to meet the requirements of countries with a highly dense population was another major issue. Tom-Aba et al. [21] assessed the roles of the design and development concepts in mobile-based contact tracing and surveyed 58 apps that were used to manage the Ebola Outbreak in West Africa between 2014 and 2015. Recently, some authors investigated the design roles in the global effort to control the transmission of COVID-19. For this, the potentials of semi-automated and fully-automated contact tracing systems were reviewed to inform the effects of automation designs on controlling COVID-19 [22]. As an early attempt, the study identified only 4036 research papers, and a small fraction of approximately 2.73% were considered for full-text screening. The 15 studies that were finally filtered, and the reviews showed that using automated contact tracing for controlling COVID-19 requires a high population uptake, between 56% and 95%. Additionally, other measures such as social distancing, lockdowns, as well as vaccine administration are typically needed alongside the control measure. Weakening data protection may be preferable to the lockdown's far-reaching limits on personal freedom and high expenses. It

might be biased to conclude that public health is more important than the users' privacy and security.

Despite recognizing that effective and timely tracing of contacts is a vital civic health measure for containing COVID-19, other contextual and environmental issues have also been the concern of the general public. For instance, a cross-sectional study was done to provide apprehensions on COVIDSafe, an Australian mobile app [23]. However, the future implications of drug and vaccine developments were the major contexts. In the face of COVID-19 variants, such as Omicron, another contextual issue behind the contact tracing strategy is when to ease restrictions for economic gains. This requires better strategies to contain COVID-19 without further lockdowns. Lockdown and testing strategies are central approaches to achieving this; Automated Contact Tracing, on the other hand, can help speed up and improve the effectiveness of strategies, as discovered in some countries. Significant logistics and capacity constraints will be required in place to improve diagnostic testing in developing countries, train personnel, and scale-up capacities for effective data analytics. As soon as the number of infected individuals has been reduced sufficiently, rapid suppression of new virus infections will be critical. In this paper, a scoping review study is carried out. We adopted the methodology specified by Arksey and Lisa [24]. This review aimed to summarize how the contact tracing strategies recently developed for COVID-19 have addressed vital issues such as the design and implementation of contact tracing systems; contextual and environmental implications of contact tracing; as well as the privacy, security, and ethical issues that have arisen in the attempts to prevent and contain the SARS-CoV-2 virus since its outbreak. We summarized the studies that focused solely on contact tracing and included an analysis of the intensity and frequency of the study in different parts of the world. Furthermore, we report how machine learning and data analytics were employed for tracing COVID-19 contacts.

## 2. Justification for the Study

Automated Contact Tracing (ACT) approaches have been diversely studied in the literature for different ends and examining different aspects; notable among those aspects are the design and implementation approaches of these tools. ACT is a major public health intervention bearing significant impact along several strata from its design to its application, deployment, and management. For instance, the technological approaches used in detecting potentially infected users, getting contacts of infected users, reporting, tracking data management, and security are some of the many potentially impacting issues that may affect the success or failure of an ACT tool in production. A broad-range investigation is necessary to consider various key strata of ACTs that may be of interest to researchers, designers, and managers involved in the design and use of these tools.

We have undertaken our review of literature in regard to technology, epidemiological intelligence and privacy issues, implementation considerations, contextual implications, privacy, security, and ethical issues. Although these cannot be considered as exclusive or as being the most important, we however selected them as core aspects of ACTs to be investigated. These yardsticks have guided the selection and examination of literature concerned with contact tracing using technology. Our discussions were also biased towards a view along these points of interest. The task of establishing existing knowledge along multiple points of interest, despite being limited to the current COVID-19 pandemic, results in a broad range of literature and possible variations in methodologies involved; these are examined primarily to extract the reported works viz-a-viz the applicable issue, in essence limiting critical evaluation of the works with general reference state-of-art in the literature. Thus, the following research questions guided this study:

- What design issues, efficiency, and implementation bottlenecks currently exist in developing and deploying COVID-19 contact tracing Apps?
- What are the differences in the approaches in terms of contextual requirements (setting, environment, population), and technological considerations in developing automatic contact tracing tools for COVID-19?

- What are the privacy concerns, security, and ethical considerations for developing a robust COVID-19 contact tracing?

Therefore, this study critically assessed how the development of ACT tools recently deployed for COVID-19 in many countries have addressed vital issues such as design and implementation matters; contextual and environmental implications; as well as the privacy, security, and ethical concerns that have arisen in the attempts to use ACTs tools to prevent and contain the spread of SARS-CoV-2 virus since its outbreak. The key contributions of this study are:

- Review of ACT studies between November 2018 and May 2021 considering implementation strategies, contextual environments, privacy, and security concerns which could provide a springboard for developing a more robust ACT especially in the African context.
- Specific country-wise analysis of the use of ACTs for containing the spread of COVID-19.
- Recommendations for researchers in developing a more dynamic and user-centric ACTs for monitoring the spread of COVID-19.
- Identification of some open and future research directions in developing and deploying a robust ACT system for containing the spread of COVID-19.

The rest of this paper is organized as follows: Section 3 presents our materials and methods. Here, a review of global COVID-19 contact tracing efforts is carried out with emphasis on design, efficiency, implementation, contexts, technology, privacy, security, and even ethical considerations. Section 4 presents the findings of our review as well as a comprehensive discussion of these findings. Lastly, the conclusion of this review as well as future research directions are presented in Section 5.

## 3. Materials and Methods

### 3.1. Databases Search Strategy and Eligibility Criteria

We formulated a search strategy that was used to explore multiple databases and find all recent and relevant research articles that focused on the development and/or validation of contact tracing apps for COVID-19 pandemic. The scoping searches were conducted primarily on PubMed (https://pubmed.ncbi.nlm.nih.gov/advanced/ Google Scholar (https://scholar.google.com, the and Association for Computing Machinery (ACM) Digital Library (https://dl.acm.org/search/ advanced databases. The search period was defined to filter all articles that were published between November 2018 and May 2021. The three databases were chosen as they provide the most complete indexes of studies on COVID-19. Additionally, the databases are relatively easier and faster ways to search across a range of study contents and publications that are considered in this review study.

### 3.2. Search Terms

The research team discussed the search terms and defined them using the two search terms in Box 1. These search terms are the most appropriate keywords that are reflected in the studies that have utmost relevance to our research questions. Aside from "COVID", "SARS-CoV-2", and "nCoV", other search terms were identified in full to exclusively avoid any potential conflicts with other terms. For instance, using ACT intended for "automated contact tracing" could extract articles related to the 1977 Infectious Diseases Act from the databases. This would make filtering rather cumbersome without having additional useful resources generated. The selection criteria were carefully designed to consider articles that contain one or multiple search terms in the title and/or abstract of the papers, whereas the further steps below are taken to ensure that the articles follow our focus.

**Box 1.** List of the search phrases used for querying the databases.

("COVID*" OR "SARS-CoV-2" OR "novel coronavirus" OR "nCoV*") AND ("automated contact tracing" OR "Mobile App* contact tracing" OR "Design Consideration" OR "Implementation Issues" OR "Design Issues" OR "Implementation Consideration" OR "Cost implication" OR "Mobile App Cost implication" OR "contact trac* APP cost consideration" OR "contact trac App scaling"; Privacy OR Users' Privacy OR Data Security OR Confidentiality in Contact Tracing OR CoVID-19 Data Integrity OR Contact Tracing Data Leakage OR Contact Tracing Intrusion Prevention OR Intrusion Detection OR Contact Tracing Data Protection OR Covid-19 privacy preservation OR privacy-preserving contact tracing OR contact tracing security architecture OR contact tracing privacy mechanism OR contact tracing ethical issues OR factors hindering adoption of contact tracing

The selection procedure is in Figure 1. We excluded preprint and review articles, whereas published studies that are observational, interventional, related to modeling, or case studies focused on manual, semi-automated, and automated contact tracing were included. From the 768 articles yielded by the database searches, we analyzed and removed all the duplicate articles that were identified from the different databases. Thus, 213 articles were filtered out reducing the articles to a total of 555 articles which were screened based on titles for study relevance. The relevance of 54 articles could not be directly determined based on their titles, thus abstracts of such papers were considered to make a decision. Further, a full-text review was carried out to make up the assessment for the remaining articles where authors lacked certainty regarding their relevance to the study objectives. A post-title filtering stage was observed and a total of 183 articles were further excluded. These included 53 articles the authors determined to be irrelevant upon screening the abstracts. Thus, 372 articles were sought for retrieval. Additionally, further screening was done to assess articles' full texts and decide on inclusion eligibility based on the criteria in Section 3.1. Finally, 117 eligible articles were downloaded for the review [1–5,8–12,14,16–19,21–120]. The data extraction procedure was done by four authors, whereas the data validity and accuracy were confirmed by the other authors. The articles were downloaded in full text and stored in a database that was shared by all of the authors. We extracted the articles' authors, titles, publication year and journal, keywords, and abstracts. Upon carrying out the full-text review, the aim and purpose of the studies were analyzed along with the contributions to knowledge. We also analyzed the studies' hypotheses and findings to comment(s) on their strengths and limitations. We were able to create information related to our study questions using the extracted data, particularly in determining the major strategies and factors that are considered for COVID-19 contact tracing systems development.

Despite early non-pharmaceutical efforts, many developed countries recognized contact tracing as an applauded approach by stepping up their research and developmental efforts toward early detection and faster containment of COVID-19. Border crossing activities remained a major route that expanded the outbreak of COVID-19. Though governments of many countries were prompted to both implement international border closures as an effective measure to contain spread of the disease, effects of the action on the economy of many countries made it unsustainable. Moreover, high rates of job loss, underemployment, and unemployment forced the re-opening of the borders, and this led to subsequent outbreak waves of COVID-19. In this section, a review of global efforts made for tracing COVID-19 contact and studying the network chaining are reported with emphasis on design, efficiency, and implementation considerations. We analyzed how mobile and edge devices were used to intensify contact tracing, and how operational efficiency was prioritized to control the virus risks in different countries.

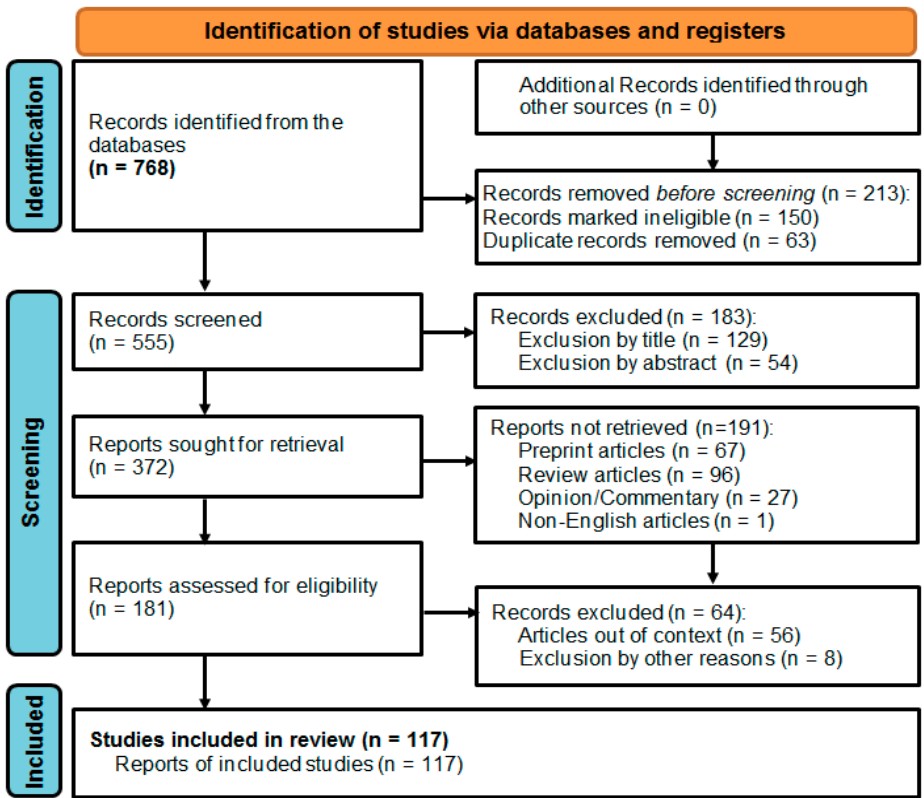

**Figure 1.** Flow diagram for the scoping review study following PRISMA guidelines [47].

*3.3. Design Consideration*

Designing contact tracing systems typically involves designing models that are used for conducting case-based source investigations in partly or fully automated ways. As a key component of controlling the transmission of infectious diseases, designing a COVID-19 contact tracing system is different from approaches for other infectious diseases such as ebola, HIV, and tuberculosis, since incidence rates rapidly outpaced the public health capacities in nearly all countries. Thus, different design considerations have been reported by many researchers. For instance, a comprehensive contact tracing system could only be good for tracing individuals susceptible to established units to a few clusters of cases; however, it may not be feasible in an environment where hundreds or thousands of cases are suddenly reported a day. Similarly, some authors had studied developing contact tracing models that are manually managed but could be adapted for optimal results based on local context and resources. When necessary, epidemiological, statistical, and related studies were done to show how population-level COVID-19 situations were managed before and after deploying contact tracing programs. Thus, the main issues considered in developing and validating contact tracing systems for establishing COVID-19 growth/spread web and transmission are discussed.

3.3.1. Technological Factors

Evidence for using fully or partly-automated contact tracing tools for containing Coronavirus was not well-known until the COVID-19 pandemic. Motivated by the limitations of using direct tracing methods such as interviews, [45] developed an innovative and practical approach that is smartphone-based, for contact tracing via relative distance estimation in ambient environment matching. A hypothesis that stemmed from using multimodal data from WiFi, acoustic sound, air pressure and magnetic field for epidemic contact tracing was evaluated experimentally. For this, the authors proposed an impeccable alternative of ensuring trustworthiness and efficiency during contact tracing with a focus on mass adoption in a wider population. Furthermore, the authors empirically verified the approach

in various real environments and the results showed 95% success in phone–phone contact tracing using multi-source data analytics. Although a low sensitivity output was obtained with unimodal data sources, authors had shown that Bluetooth-only data is also the best unimodal performer. Ultimately, the study proposed a system that can be used to be reliably used for contact tracing with higher multimodality information levels for vital public health decision-making. Yet, further studies are needed to effectively differentiate between true contact tracing in a shared-air space and segregating false ones with instances of thick walls. To deal with the COVID-19 lockout and the economic crisis, a smartphone-enabled method was devised. Spachos and Plataniotis [44] designed a classification system that is based on the user's proximity sensing and privacy preserved with a communication protocol. This shows signatures broadcasted and stored locally on a user's smartphone can enhance contact tracing. The model has been facilitated with experimental data which is publicly available for further validation.

Design intelligence has also been considered for more suitable implementation in some studies. For instance, Vedaei et al. [16] and Fernández-Caramés et al. [121] presented potential applications of IoT (Internet of Things)-based healthcare systems for physical distance monitoring during pandemic situations and carried out an evaluation study with the COVID-19 scenario. Health parameters such as cough rate, respiration rate, body temperature, and even blood oxygen saturation are tracked by the systems, which are then used to represent a user's health status. The IoT system could assist users in monitoring daily activities and warn them about exposure to the COVID-19 virus. A limitation is that the app prevents COVID-19 with a notification module built to remind users to keep a physical distance of 2 m. Nakamoto et al. [61] evaluated a COVID-19 contact-confirming app that is designed on Bluetooth technology and mobile contacts for containing the spread of COVID-19 in Japan. An initial peer-to-peer COVID-19 contact tracing app was carried out, using a mobile app-based implementation shows the app could prioritize users' privacy from potential attackers and public authorities, and enhance balancing between loads of excessive pressure on the healthcare system during the pandemic. A simulation study with the app shows that higher study participation is needed (in Japan) for effective control of the virus. Meanwhile, an extended study in Nakamoto et al. [122] shows that integrating QR code–based operation in the contact tracing framework could aid sustainable containment of the virus. For the latter, further evaluation reported shows that the approach assisted in returning activities of the study area to normal timely. All the same, while using a symptom-based QR code to enhance quick tracking and societal response to containing COVID-19 spread, the authors show that dual QR codes might be used for national contact tracing, exposure risk self-triage, healthcare appointments, and self-update of health status, and could be formally treated as electronic certificates of persons' health status, without retrieving location data of the users. The study had considered only two-color QR codes and, thus, analyses of the flexibility and sensitivity of the model are difficult to observe. Furthermore, the authors did not provide how to address containment in real time once a sensitive situation is perceived in the system.

Similarly, Ryan et al. [86] developed a model based on Bluetooth or pairing devices data sources for COVID-19 contact tracing in crowded meetings and high-density activities. The study aimed to address the potential for widespread COVID-19 transmission in mass gatherings but the authors just presumed that Bluetooth or similar wearable devices can be made available at crowded meetings for assessing the participant contacts and did not identify an actual model design. For lockdown management, Maghdid and Ghafoor [123] used an unsupervised machine learning technique (k-means). The study included the development of a dashboard panel that can be used to advise health officials on how to properly return individuals to their normal lives in a specific location. A weak point in this study is tracking users' location information without privacy protection. Furthermore, the study only considered Android implementation for the contact tracing application. Smart contact tracing systems that can automatically discover possible infected individual

connections require the use of smartphone built-in Bluetooth low-energy signals and machine learning classifiers.

### 3.3.2. Epidemiological Intelligence and Privacy Issues

Three ethical considerations can be considered in a blueprint for the ethical evaluation of contact tracing systems. For instance, while analyzing the technical blueprint for the ethical review of contact tracing apps, Lanzing [67] considered data monopolists, privacy, and coercion-based concerns. The author explored the two scenarios that start with envisioning and critically evaluating contact tracing app-based built on anonymity-based privacy concepts for individual rights. The second scenario depicts and critiques an app that appropriately solves privacy issues while being aided by data monopolists such as Google and Apple. In the last scenario, a privacy-friendly, independently made contact tracing program is forcefully installed and used. Kim and Chung [60] presented a safe COVID-19 contact tracing app for protecting privacy breaches by using functional encryption methods based on spatio-temporal trajectory data. Coercion through social isolation and limited societal involvement is the main source of concern. Hypothetically, epidemic simulators can model the effects of arbitrary population testing as well as contact tracing. Thus, Kuzdeuov et al. [42] designed a particle-based COVID-19 simulator model that idealizes each individual as a particle for contact tracing purposes over velocity, position, and even epidemic states on a 2D map. The proposed model was validated by analyzing epidemic situations based on susceptible, exposed, infectious, and removed (SEIR) statutes for appropriate contact tracing. The study demonstrated that the particle-based COVID-19 simulator could achieve a 72% mortality reduction; and as a result, it is capable of aiding modeling of intervention measures, random testing, and contact tracing, for epidemic mitigation and suppression. The evaluation study was carried out by comparing the proposed model to a compartmental SEIR model and this reveals that a more realistic or accurate simulation of disease progression can be achieved, allowing for the development of suppression and mitigation techniques.

Yu [57] proposed a secured contact tracing framework using a risk infection model that preserves users' data, thereby increasing its adoption among health agencies. Findings indicate that the approach adopted performs better than other similar approaches. The app's strength is in proposing the ideal local test locations for a person who may have been exposed to the virus, but by altering the disease vector, risk model, and environmental influences, it may also be used for other contagious diseases. Furthermore, [48] also wanted to create a realistic contact tracing system that did not compromise privacy or usefulness. They employed design principles based on seven privacy considerations to detect privacy flaws in commonly used technological solutions and suggest a new way of contact tracing that protects individual privacy. Individuals' desire to accept the technology will be aided by the authors' adoption of solid privacy norms. It also ensures that organizations comply with privacy laws and regulations, such as GDPR, while keeping consumers and employees secure. Garg et al. [31] designed a novel privacy anonymous IoT model based on RFID proof-of-concept and blockchain's trust-aware decentralization for anonymity preservation in COVID-19 contact tracing. The model utilizes trust-aware decentralization in blockchain technology for on-chain data storage and retrieving; meanwhile, the RFID-based conceptual model was simulated for validation purposes. The simulation results show it could achieve milli-second model deployment for smart contracts. It was that the model is presumably going to take about 25 s on the Ethereum public blockchain for realistic scenarios. Related studies were reported in Hao et al. [33] and Klaine et al. [41]. The effectiveness of the app was not tested on real-life scenarios of providing safe and unsafe location information to users. The former shows the capability of resource allocation, energy consumption, security, and privacy issues compared with existing solutions. Kassaye et al. [39] shows that this technology however fails to capture COVID-19 symptoms.

*3.4. Implementation Considerations*

Contact tracing works are majorly focused on containing the outbreak on time. Idrees et al. [124] considered the different techniques used in COVID-19 contact tracing app development. Design consideration analysis should not preclude integrating robust response implementation testing that can enhance timely deployment and usability. Deployment of smartphone-based apps is vital for promptly implementing community-level mitigation measures such as physical distancing in public places, limited outings, avoiding large gatherings, wearing face masks, washing hands often, and so on. With the existing design considerations, studies on contact tracing systems have been implemented with peer-to-peer and labor-intensive mechanisms.

### 3.4.1. Peer-to-Peer Contact Tracing

Yasaka et al. [15] describes a privacy-preserving peer-to-peer contact tracing smartphone software that respects user privacy and conducts a novel kind of contact tracing using an anonymized graph of interpersonal contacts. The authors created a proof-of-concept smartphone app and validated the model using computer-based simulation. The research shows how the proposal affects pandemic outbreak trajectories at various acceptance rates. It allows users to construct "checkpoints" for contact tracing, assess their risk level based on previous encounters, and anonymously self-report a positive status to their peer network. The established contact tracing app's acceptability is a serious restriction. The use of location-based traffic detection algorithms has been discovered as a reliable method for determining user location at points of contact. This procedure would however present some potential privacy concerns as study validation shows users may not be comfortable having the app track their real-time location. The implementation also lacked ways for ensuring that confirmed diagnoses are reported via the app. Users who became COVID-19 victims could be prickly in doing self-identification through the app despite being anonymous.

### 3.4.2. Labour-Intensive Contact Tracing

In Ye et al. [92], a health informatics system to better comprehend the Chinese health community's actions during the COVID-19 outbreak was developed. Mobile and web-based services such as WeChat and QR codes were employed whereas big data analytical modeling was developed for digital contact tracing and epidemic prediction. In reality, health informatics played a critical role in reacting to the COVID-19 epidemic. Wang et al. [53] also integrated a contact tracing app into WeChat to monitor and locate suspected patients, using spatiotemporal data and GPS. This data could provide users with a future epidemic trend prediction tool. Blockchain technology was however proposed for future work to mitigate the limitations of GPS, as well as combat privacy issues. Yap et al. [56] developed a web-based symptom monitoring app that enabled users to keep daily personal records of health status and movements. This is to supplement government management strategies for COVID-19. Ye and Lee [70] present a comprehensive mobile device-based COVID-19 contact tracing system through Proximity Sensing. The work is motivated by seeing society return to full normalcy through its deployment and adoptions. They addressed both intermediate and long-term solutions to some of the issues faced by the current generation of proximity-sensing technologies such as Bluetooth Low Energy (BLE), which is utilized by phones. Findings from probabilistic models reveal that for a solution to be effective, it must be adopted at a high rate, as much as 90–95 percent (depending on disease parameters).

### 3.4.3. Adoption of New Digital Technologies

Complementary contact tracing that could enhance the management strategies for containing pandemic outbreaks during early future waves of infectious diseases, e.g., COVID-19 can be aptly done. Response to the COVID-19 epidemic has also been investigated from digital technology and health informatics perspectives [125]. From the

perspective of case detection, Jian et al. [59] demonstrated how digital tools could help prompt contact tracing and management of COVID-19. A centralized contact tracing system was created to facilitate data integration, cross-jurisdictional cooperation, and contact monitoring by following up on contacts' health conditions. Traditional contact tracing approaches and symptom-based tracking are not effective in aiding contact management in public. Polenta et al. [49] introduced "the BubbleBox System", an IoT Approach to Contact Tracing. The goal of the project was to define the BubbleBox architecture, demonstrate a prototype implementation, and explore its advantages and disadvantages, as well as privacy problems. The experiment did, however, use a prototype to evaluate the distance estimation, as mentioned by the authors. Usability tests with the right design are required to completely validate the BubbleBox device, for instance, using state-of-the-art metrics such as the System Usability Scale (SUS) or the Usability Indicators for Consumer Experience to assess the user experience with the bracelet and the system. In [126], a model for COVID-19 virus transmission in Boston, analyzing the progression of the pandemic and the efficacy of social-distancing interventions was designed. Results demonstrate the effectiveness of isolation and stay-at-home policies, evident in the reduction of new cases and a controllable effect on healthcare. However, considering the effect of the study in aiding decision-making for the government, the results were based largely on assumptions (design). Another study was carried out by Cheng and Hao, [27] to design a contact tracing app that helps in tracing suspected cases of COVID-19, especially for developing countries with limited access to good healthcare. It was developed as a plugin for a smartphone app. Results show that the platform is more secure compared to other contact tracing apps. This is due to a non-centralized server adopted that considers users' privacy.

### 3.5. Efficiency Considerations

The effectiveness of app-based contact tracing is a factor to consider. To study the accuracy and effects of contact tracing on containing COVID-19, there is a need for reliable tracing technology, and contacts must be traced promptly. In addition, the mobile units need to process a substantial amount of the target population data through contact tracing applications with strict requirements. Hernández-Orallo et al. [58] showed that an epidemic model can be deployed for efficient contact tracing. Design and characterization of the model's efficiency in precisely tracing contacts were demonstrated with smartphones and edge units. Authors derived stochastic and deterministic models as contact tracing mechanisms for evaluating the effectiveness of smartphones in containing the spread of infectious diseases. A study by Park et al. [65] also examined the effect of digital contact tracing on COVID-19. The base hypothesis was proposed that civilian participation, government transparency, and high testing capacity determine the success of COVID-19 tracing. These studies reported that contact tracing operations based on GPS data kept decentralized and encrypted could augment traditional contact tracing methods for increased accuracy, and more strict legal jurisdictions should be set. Though it is known that digital contact tracing allows for quick and easy detection of active cases and potential carriers, the authors have not demonstrated the design consideration and implementation concerns needed for the actual application of the technology. Cost considerations were analyzed as the number of people quarantined in the studies.

In Almagor and Picascia [64], the use of agent-based models to explore a COVID-19 contact tracing app's efficacy was introduced. The goal of this effort is to contribute to a better understanding of the complexities involved in the interplay between COVID-19 circulation and the proposed mitigations. On an urban scale, the model mimics the spread of COVID19 in a population of agents. To analyze the impact of the pandemic, the authors looked at how different adoption rates of the contact tracing app, varying degrees of testing capacity, and behavioral factors interacted. The findings imply that a contact tracing app, when combined with enough testing capacity or a testing policy that prioritizes symptomatic cases, can significantly reduce infection rates in the population. The prevalence of infection reduces as the user rate rises. When symptomatic situations

are not prioritized for testing, app users' demand for testing can significantly increase, which, if not met with enough supply, can make the app ineffective. The findings suggest that when more people use the CTA, the virus's transmission is decreased, and so the benefits are transmitted to a larger population. The CTA, in theory, provides efficiency and cost savings that can be used to supplement and expand standard manual contact tracing approaches. A study by Song et al. [95] presents automated medical-grade COVID-19 systems for promoting early symptom identification and strict sanitary standards (both hardware and software). Body temperature measurement, sanitary compliance evaluation, and personalized person-to-person tracking are all included in the system. The system encapsulates all of the aforementioned functionality in both hardware and software and is further bolstered with preliminary "real-time locating system" (RTLS) data capture, allowing for post-symptom detection, and person-to-person contact identification to assess potential infection vectors and mitigate further spread through smart quarantine.

Epidemic containment with digital contact tracing as one of the first models for assessing SARS-CoV-2 transmission was proposed in [1]. A mobile phone app hypothesis was noticed that might indicate how constrained resources must be allocated among diverse intervention tactics for the most effective control. The authors did this by estimating the contributions of various transmission paths, developing a renewal formulation, and calculating the speed of contact tracing necessary to end the epidemic analytically. Furthermore, a new model for measuring infectiousness to the fundamental reproductive number R0 was established to evaluate the role of various transmission channels in epidemic containment via contact tracing. By studying 40 well-characterized source-recipient pairs, as well as estimating the distribution of generation times (the time from infection to transmission), the study demonstrates how to parameterize the model. A limitation observed is that the initial model was observed in early epidemic data obtained in China only. Braun et al. [26] predicted the effects of interventional procedures to regulate the COVID-19 pandemic using a modified Bateman SIZE algorithm. The daily number of infectious persons that peaked in 18 months could be prevented with a vaccine that has 50% efficacy. Thus, a combination of effective vaccination and diagnostic tests along with the usage of Corona-Warn-App with quarantine might effectively control the spread of the coronavirus contagion. Recently, some authors have developed a crowdsourcing model, with large-scale evaluation, for collecting contact tracing data to encourage users in contributing more contact tracing data [52]. The latter is a scalable simulation study that showed how publicly contributed data could enhance contact tracing. The study shows higher performance over benchmark data collection methods, with the former capable of 13% user participation improvement for different scenarios. Although the evaluation was done with real-world data, the model validation reports are from a simulation study that necessitates further real-life experimental works.

### 3.6. Contextual Implications of Digital Contact Tracing

In curtailing the spread of COVID-19, several authors have proposed various forms of experimental designs for digital contact tracing. Related studies are in evaluating the effects of some key factors such as culture, environment, and population on the acceptability, usability, and effectiveness of digital tracing of contacts throughout the epidemic.

In this subsection, relevant studies on COVID-19 contact tracing are examined to investigate how effective digital contact tracing systems respond to contextual and environmental implications during the COVID-19 contagion. Studies have been conducted to investigate the spread of COVID-19 and the effectiveness of non-pharmaceutical interventions and validate contact tracing efforts. Currently, the most effective way to contain the COVID-19 epidemic leverages digital contact tracing [52]. However, another contextual issue is that it requires monitoring people's daily life and data about the contacts they make to decide and who needs to be isolated for every positive COVID-19 case detected. Some governments have identified rewarding systems as a positive way of getting real-time analysis of COVID-19 situations in their areas rather than investing huge human and

physical resources to combat it. In some countries, leading efforts to develop mobile apps for contact tracing initiated voluntary participation and guaranteed guardrails to protect confidentiality [33]. Nevertheless, lawmakers are trying to weigh the risks involved in contact tracing against rewards. Wang et al. [52] studied the procedure of gathering contact tracing data from a crowdsourcing perspective to encourage users to contribute additional data and develop the "CovidCrowd" incentive algorithm. The researchers structured the problem as a Stackelberg game and demonstrated that given a fixed reward amount, any user may reach a Nash equilibrium. Then "CovidCrowd" calculates the best reward value that will maximize the system's utility. Finally, they ran a large-scale simulation with thousands of users, as well as a real-world dataset review. Both results reveal that "CovidCrowd" surpasses the benchmarks; for example, for all evaluation situations, the user participation level is improved by at least 13.2%. Users choose their response tactics based on the total reward value, which the contact tracing system reimburses them for their privacy and data processing costs. "CovidCrowd" outperforms other algorithms because it uses the Nash equilibrium of the Stackelberg game to determine the optimum response strategy for both the contact tracing system and the user. Despite these feats, [91] designed an IoT-based approach for contact tracing, where virtual home numbers are issued and attached to real households in the community. This is propagated to the devices in the smart home setting and afterward, the ring signature is transmitted to a quarantine smart contract (via a smart gateway). Subsequently, future events such as an emergency can be sent by the gateway to pandemic control along with the virtual house number. A study by Ferretti et al [1] examined the important factors of the COVID-19 epidemic spread to evaluate the role of various spread conduits and establish case isolation and contact tracing criteria. The scientists found that viral spread is too quick to be controlled by manual or conventional contact tracing and that nonetheless, this technique may be made faster, more efficient, and scaled up. The approach is to construct a contact tracing program that creates a temporary record of individuals' proximity events and warns of recent close contacts of identified cases, prompting them to self-isolate. A study by [68] presented a chatbot and a symptom-to-disease digital health assistant capable of distinguishing over 20,000 diseases. In Cheng et al. [27], a central agent system that receives messages from infected persons and broadcasts them to close contacts of uninfected persons was proposed. The list of close contacts, provided by the user, receives a code; thereafter, the user needs to visit a website where they can view the message and the accompanying health advisory of steps to take. It involves a crowdsourced exposure database containing a trajectory as well as trajectory information.

### 3.6.1. Location Tracking and Analysis

This tool can be used to simulate physical separation measures by omitting the contribution of mixing patterns in specific areas to examine the influence on disease transmission and provide guidance to policymakers. A study by [55] built "SOCRATES", an online tool that assesses COVID-19 mitigation measures by exploiting a social contact/connection data sharing program. The researchers used internet repositories and formatting rules to manage data sharing of published social interaction surveys. Weighted social-connection matrices, next-generation matrices, comparative incidence, and R0 were used to analyze human social interaction data. They took into account location-specific physical distancing measures, such as school closures or work, and measured their impact on transmission dynamics.

At an Infectious Disease Center in Singapore, Ho et al. [36] evaluated the application of a real-time locating system for contact tracing of healthcare workers during the COVID-19 Pandemic. This study used the RTLS and a review of the "electronic medical record" (EMR) at the designated hospital for COVID-19 response in Singapore to compare the effectiveness of contact tracing during the COVID-19 pandemic. The authors of the previous study verified the web tool using solely social contact data and showed that physical distance had a significant influence on minimizing COVID-19 transmission. Meanwhile, Ho et al. [36]

provides a more practical perspective on the utility of EMR- and RTLS-based contact tracing approaches in "real-world" situations. To ensure reliable data collecting, the study team used a scoping and consistent methodology to execute contact tracing by using two methods and conducting telephone interviews with health care workers. In general, the outcomes of these studies add to the scant data on the efficiency of RFID technology against traditional contact tracking methods. The majority of RTLS research is limited to the measuring of human interaction length in health care settings. The authors employed location-based tracking to locate contacts for the RTLS-based contact tracing and did not examine the contacts' closeness.

According to Shelby et al. [127], contact tracing can be viewed as a core component of community health response to emerging infectious ailments. Lack of suitable knowledge on the contextual implications of implementing COVID-19contact tracing systems added to the pandemic's dynamics. For instance, region-level and individual-level predictions are critical to properly contain the disease earlier and prepare for the new variant and future waves that are still seen today. In response, Jian et al. [59] investigated contact tracing using digital aid in Taiwan's COVID-19 epidemic response. Traditional contact tracing procedures augmented with symptom-tracking and contact information management were shown to be useful in assisting public health personnel with high efficiency. To assist with data linkage, cross-jurisdictional synchronization, and follow-up on contacts' health conditions, a centrally managed contact tracing system was established. They demonstrated how digital tools aid in the contact tracing and management of COVID-19 cases, as well as evaluating system performance by measuring the time it takes from case discovery to contact monitoring. The rate of health status updates through self-reporting grew from 22.5 percent to 61.5 percent after applying the strategy of self-reporting via automatic text messages as well as web apps. The researchers stated that a pillar of methods for controlling outbreaks amid de-escalation or earlier in the next phase of the COVID-19 pandemic would remain extensive contact tracing and management using complementary technology.

Signals from mobile phones and gadgets of COVID-19 victims can be utilized for contact tracing analysis as well. Leading technology developers such as Google and Apple proposed to launch mobile apps that can enable phone tracking-based Bluetooth signals emission [33]. With this, the phones of every COVID-19 victim can be used to stratify people that are at risk of catching COVID-19 and further classify persons that may be suggested for further testing and diagnostic steps. A study by [128] explores how detection ranges and usage stops affect the effectiveness of digital contact tracing. They give a quick rundown of their simulation process. For the simulations, they used earlier epidemiologic research and evidence to determine where illnesses are most likely to occur, and they used four different types of settings: residences, educational institutions, workplaces, and supermarkets. The SEIR model, which asserts that a person can be in one of four stages: susceptible, exposed, infectious, or recovered, was used to create the epidemic in the simulation. Initially, practically everyone is vulnerable, and only a few people are exposed to contagious diseases. After some time, exposed individuals move into an infectious state, in which they can spread the disease to other people. An infectious person recovers and becomes immune to the disease after some time. Their findings show that regulators should tailor digital contact tracing programs to a society's behavioral traits. Based on this, the closeness detection limit should at the very least cover the spread of disease, and in some circumstances be considerably wider. Their findings imply that the most promising CT solutions for many cases are based on a PDR that essentially correlates to the infection spectrum. This conclusion, however, is contingent on several factors, including the initial acceptance rate and likely usage stops. They also highlighted that effective CT for COVID-19, in a broader context, highlights the power of digital technologies in addressing future difficulties for global health care systems.

### 3.6.2. Location-Based Social Network Tracking Technique

The major goal of contact tracing systems for disease control is to provide users with personalized status updates about their vulnerability to disease within a geographical location. Considering the orientation and location of users provide more precise status information. A location-based social network (LBSN), also referred to as a geo-social network, is a type of social networking platform where geographical services are integrated with traditional social networks [78]. The geographical services feature enables new social dynamics, including hidden patterns derived from visits of users to the same or similar locations, in addition to knowledge of common interests, activities, and behaviors inferred from the set of places visited by a person and the location-tagged data generated during these visits. User preference, social influence, and geographical influence are three aspects responsible for users' patterns and check-in activities. The user's pattern is derived from user-based collaborative filtering and social influence is based on structural patterns hidden in the nodes of a friendship network [76].

LBSN has been considered to study to understand human mobility and behaviour; for example, a location-based orientation-aware recommender system that applies contemporary user contextual information towards more personalized direction-pointing recommendations in a stampede [89]. This recommendation system can inform users about movement directions to avoid disease locations. Additionally, investigating network structures created in a location to explore the evolving history and scientific collaboration patterns of contact tracing have been explored to understand the connection between people [81]. The relationship pattern in a network is evaluated on network cooperative closeness, average shortest path length, average clustering coefficient, and community detection. In addition to considering the location and connection, the strength of the connecting node in LBSN was explored to implement friendship prediction [78]. The prediction method considers the hidden pattern in vertices between friendship nodes and changes in these patterns as the user's preferences change through interactions. Similarly, the social-spatial influence and social-temporal influence in LBSN were synthetically analyzed to reveal check-in movement behaviors of users are affected by their social friendships both from spatial and temporal dimensions [95]. The Check-in feature in LBSN enables people to share real-life activities and location information. The analysis of historical records of check-ins to predict future locations is essential to identify possible places visited by infected persons and areas for disease prevention and containment [77].

Despite the benefits of the LBSN structural pattern, LBSN is vulnerable to security threats that could target both the users and resources. An adversary may violate a user's location privacy in two ways: (i) based on the user's location information contained in the LBS query payload and (ii) by inferring a user's geographical location based on the device's IP address. Therefore, an implementation of a context-aware privacy-preserving location-based services system with integrated protection for both data privacy and communication anonymity was proposed [75]. Similarly, another consideration was a proposed blockchain-based lightweight certificate authority for privacy-preserving location-based service [84,88]. A threshold proxy signature is issued by a coalition of certification authorities playing the role of authorized nodes in a consortium blockchain.

### 3.6.3. Real-Time Location System Development

Real-time location systems (RTLS) have been widely used for tracking and automatic identification of humans and objects in many domains. Usually, the technology is used in manufacturing industries for locating in a time-sensitive manner such that managers can keep up-to-date information of tagged pieces. Such ideas have also been used in biomedical engineering such as attaching a tag to a patient in the hospital for proper health monitoring, vital sensing, and many more areas. Digital and automated contact tracing has been largely based on mobile phone apps that have been developed to monitor and assess the potential drivers of COVID-19. A study by Grantz et al. [129] indicated that mobile phone data could aid the analysis of COVID-19 pandemic epidemiology, and

Nwawudu et al. [46] proposed the development of an instantaneous COVID-19 contagion contact tracing application using a mobile phone to aid in the gathering and transmission of adverse occurrences related to the surveillance, and managing of the COVID-19 pandemic around the world, particularly in rural areas. The centralized datastore and Web server, remote access to the database from just about any Internet-connected pc, telephone audio computer-aided personal interviewing, voice messages, and SMS-based communications to and from the server via mobile phones are the five architectural elements of this proposed system. This proposal depicts how a full COVID-19 surveillance system can be constructed using mobile phones with the right communications technology partner. COVID-19 contact tracing can be developed using mobile phones as a viable technique for collecting and reporting data in real time, according to the research. However, they noted the following drawbacks of employing mobile phones to aid in real-time COVID-19 Pandemic contact tracing: social concerns about data breaches, the provision of incorrect medical advice based on self-reported symptoms, and the scoping exclusion of some parts of society who are unable to access the system.

Regarding containment of the virus spread, such as with the COVID-19 disease, El-Rashidy et al. [130] presented an end-to-end deep learning model. This model's primary goal is to close the gap between present technologies and healthcare systems. To create a comprehensive and extensive model for disease discovery and monitoring, the wireless body area network, cloud/fog computing, as well as clinical decision support systems was merged. Meanwhile, operational costs such as the power consumption of the wireless sensors and highly consuming capacity during sensing are major challenges in the monitoring system. Similarly, through the development and successful 72-h deployment of a Hybrid System for COVID-19 in Hubei (China), [30] demonstrated how modern medical informatics technologies have enabled effective pandemic containment. The system was developed to collect, integrate, standardize, and analyze COVID-19-related data from a variety of sources, including diagnostic labs, a case reporting system, mobile social media, and even "electronic medical records" (EMRs). Based on geo-location data via mobile-cellular networks, Rahman et al. [131] proposed a computerized contact tracing strategy for reducing COVID-19 dissemination. In contrast to BLE-based tracking strategies, the researchers suggested a contact tracing method based on cellular SIM card geo-location data. A confirmed COVID-19 patient's mobile number was provided to the corresponding mobile operator in the suggested model to locate the patient's mobility information for the previous 7 days. The operator tracked the movement using the geolocation data. The operator employed the geo-location-based tracing technique by receiving position data straight from the base station to detect the likely infected persons in the suggested model and, therefore, no Bluetooth/Wi-Fi/NFC capable mobile phone is required.

### 3.6.4. Technological Considerations in Contact Tracing Apps

This strategy prevents panic from spreading among individuals because it does not require constant warnings about the risk of infection. The suggested strategy relies on mobile users' geo-location data obtained directly from the carriers. To build contact tracing systems for COVID-19, modern methodologies have been proposed. El-Rashidy et al. [130] proposed real-time tracking system is based on a multi-layer deep learning framework proposed with three different layers; identified as the cloud layer, the patient layer, as well as the hospital layer. Patients are tracked in the first layer using a set of wearable sensors and even a mobile app, whereas in the cloud layer, a fog network architecture is presented to handle storage and data transmission difficulties. Based on patients' X-ray pictures and transfer learning in the hospital layer, the authors developed a convolutional neural network-based deep convolutional neural network for COVID-19 identification. Unfortunately, not all smart devices can transmit aggregated vital signs without the need for human involvement. Furthermore, COVID-19 cases were identified based on an initial score that might or might not be reliable in all cases. Additionally, despite the benefits of fog nodes, such as privacy, security, and productivity, fog nodes may add to the network

infrastructure's complexity. It also necessitated greater upkeep for the dispersed storage nodes. IoT has been utilized as a technological consideration for developing COVID-19 contact tracing systems. For instance, Thangamani et al. [132] developed an automated method for detecting coronavirus contacts using IoT devices. The system's design and execution are based on a prediction technique for reporting patients' healthcare risks based on variables including temperature, humidity, and blood pressure. This includes an architecture of IoT devices that connects to share a person's data read through sensors and is shared with the doctors via a mobile, PDA, or a related computer to predict the person's COVID-19 status. As a result, the systems may detect all symptomatic and asymptomatic patients, assisting in disease prediction. However, this comes with a lot of contextual issues regarding users' privacy and data security.

Meanwhile, Ng et al. [44] also introduced using a smartphone's BLE for a smart contact tracing system, a safe alternative to manual contact tracing. The system detects Bluetooth signals and uses a machine learning classifier to assess users' contact profiles accurately and fast. This technology has a non-connectable feature that is used to broadcast signature packets as users get to a public space. Thus, when a user is confirmed to be infected by public health authorities, the broadcasted and observed signature saved in the user's smartphone is uploaded to a secured signature database. The technique is a notable solution for contact tracing during epidemic control and prevention, according to experimental data. Although overall contact tracing performance improves while maintaining user privacy, such systems rely on a centralized database, which can lead to security vulnerabilities such as "man in the middle" (MITM), "distributed denial of service" (DDoS), SQL injection, and so on. A study by Hoffman et al. [133] investigated the design of "Zwaai" (Dutch for "wave"), a QR code-based tracking app. The app was designed to be a viable alternative to the widely used Bluetooth and GPS-based solutions. "Zwaai" was developed by various members of the Radboud University Nijmegen's Interdisciplinary Hub for Security, Privacy, and Data Governance (iHub). For contact tracing, "Zwaai" uses a QR code-based architecture. Each user's app generates a constantly refreshed QR code and is capable of scanning other users' QR codes as well; the QR code is what keeps track of the kind of interactions a particular app user has had, albeit in a privacy-protected manner because no unique identifier is assigned to any specific user or their app. This study argues that by evaluating these designers' choices, QR code infrastructures can expose several ethical or political seams, such as networked impermanence, in which seamless protocols such as Bluetooth explicitly seek to avoid. This may otherwise go unseen by existing ethical standards.

### 3.6.5. Artificial Intelligence and Machine Learning in Contact Tracing

Artificial intelligence (AI) and related machine learning (ML) technologies are increasingly prevalent in diagnosis and treatment recommendations, pandemic control, and user tracking to control the spread of disease [117,119]. ML is an important aspect of AI, which involves training models with data to "learn" and predict similar patterns. An example of ML implementation is the prediction of prospective treatment protocols that tend to succeed in a patient based on patient attributes and the treatment context [95]. There are several research studies suggesting that AI can perform as well as or outperform humans at important healthcare tasks [82]. The benefits of AI techniques have been highlighted in the implementation of COVID-19 vaccine development [83], screening viral and COVID-19 [79], Predicting COVID-19 Malignant Progression [90], diagnosis and identifying and managing of COVID-19 patients [73]. The evaluation of the performance of AI in COVID-19 can be seen in the COVID-19 genome sequences [71]. The sequential pattern mining (SPM) technique was applied to the corpus of COVID-19 genome sequences to unveil interesting hidden patterns. Frequent patterns of nucleotide bases were observed from the mining, and relationships were established. In addition, sequence prediction models were implemented to evaluate if nucleotide base(s) can be predicted from previous ones. Finally, for mutation analysis, an algorithm is designed to find the locations in the genome sequences where the nucleotide bases are changed and to evaluate the mutation rate.

AI has an important role to play in epidemic forecasting in the future. In the form of machine learning, it is the main capability behind the development of tracking, monitoring, and recommendation to advance disease outbreak management. Although successful efforts have been achieved in implementing AI towards curbing the COVID-19 pandemic, it is expected that AI will ultimately master the pre-pandemic and post-pandemic domains as well. Furthermore, the challenge to AI in this domain is not whether the technologies will be capable to deliver, but rather their acceptance either due to lack of facilities to support its operation or lack of interest to use it. AI systems must be approved by regulators, require frequent updates and improvements, and be integrated with the familiar system, which may require learning to use by clinicians and patients to achieve wide acceptance. These challenges can be defeated with time as the technology is implemented in more domains. Therefore, it is expected that there will be more use of AI within the shortest possible time. Moreover, it is obvious that on a large scale, AI will not replace health workers but will provide great support for healthcare provided to patients. Over time, health workers will focus on roles relating to human skills and emotions.

### 3.7. Privacy, Security, and Ethical Issues

Although contact tracing or tracing is a very potent tool in the fight against COVID-19 or even other epidemics of that nature, there is an explosive increase in concerns for the privacy of individuals whose information is gathered directly or indirectly, insofar that the nature of the information collected for automated contact tracing is often personal or private data, and, thus, opens the way for further discussion and social studies about users' tolerance and adoption. Usually, skepticism about making intelligent inferences about the users' private and social lives, location per time, and their health status from acquired data is another sub-topic that is considered. In the realms of security and privacy, for example, location is known to tell a lot about a person's identity [87]. Thus, researchers, scientists, medical organisations, countries, and other stakeholders have proposed and even developed varying approaches. Based on security and privacy concerns related to contact tracing, we herein classify the studies available on COVID-19 contact tracing into three. These approaches can be broadly classified under two or three major models which are centralized, decentralized, as well as hybrid (centralized-decentralized) models. To amass and analyze contact data, share secure info, and notify users about potential COVID-19 exposure, centralized or non-centralized approaches have been proposed. However, the two methods differ in terms of anonymity and techniques to ensure that data providers' privacy is protected. Meanwhile, hybrid approaches have also been developed for a well-balanced trade-off during COVID-19 contact tracing. We further investigate how each approach has been adopted for COVID-19 containment and prevention during the pandemic.

### 3.7.1. Privacy in the Centralized Contract Tracing Model

Data handling in the centralized contact tracing model includes the gathering, integration, as well as, sharing of data between instituted authorities (such as the government or health ministries) and selected persons. In this model, the privacy of users, who share their information with the government/health institutions is not guaranteed. Such users would have to just trust that the government will keep their information secret. A study by Simmhan et al. [62] presented "GoCoronaGo" (GCG), a digital/automated contact tracing software for universities that aims to overcome privacy concerns. More specifically, the design concept, architecture, and experiences gained while integrating the "GoCoronaGo" app as a component of an experiment at the "Indian Institute of Science" (IISc) were investigated in this study. It also looks at how to improve the value of electronic contact tracing by addressing the obstacles and opportunities. The "GoCoronaGo" (GCG) contact tracing platform includes a smartphone application as well as data gathering, management, and analysis backend services. This software is intended for COVID-19 management and operation within a facility, as well as an investigative project authorized by the "Institute

Human Ethics Committee" (IHEC). This system is unique in that it continuously collects device contact trace info related to just about everyone into a central datastore. All app users' proximity data is utilized to create a temporal contact graph, in which the vertices represent devices whereas the edges designate the closeness of devices for a particular time interval. François et al. [30] proposed a privacy-preserving method for gathering information about a social graph anonymously. Bluetooth-enabled "contact tracing apps," which collect information about user proximity to estimate the risk of COVID-19 spreading among them, are a common implementation of this protocol. The key contribution of this study lies in the ability of a central server to build an anonymized graph of user interactions. This graph provides the central authority with information about the virus's spread and allows the authority to run prediction models on it while maintaining user privacy. The most important technical tool used is an accumulator scheme which can keep track of the credentials.

In Kim and Chung [60], a privacy-preserving contact tracing technique based on a Spatio-temporal trajectory that can be used in several quarantine systems for COVID-19 was developed. The adoption and use of a functional encryption technique allows the suspected contacts of infected patients to be retrieved without compromising privacy. This study used optimization algorithms and parallel processing to improve processing efficiency. A visualization system was also established to display the contact tracing workflow. If enough CPU cores are available, the proposed system can easily handle a considerably larger dataset. Because the decryption computation is separately executable, both multi-core and distributed computing can be used to speed up query handling. The authors suggested that support for multi-source environments could be considered in future studies. To protect multi-source datasets from other data sources, each dataset could be encrypted with a distinct key. The Singaporean "OpenTrace" app was evaluated by Leith and Farrell [28], to determine its influence on user privacy. The "OpenTrace" app, for example, uses Google's Firebase service user data storage/handling. This means that the data transmitted from the app is handled by two main parties: Google and the health agencies that operate the "OpenTrace" app. The usage of Firebase Analytics telemetry by OpenTrace means that the data transmitted by OpenTrace could potentially allow Google to track the (IP-based) location of user phones over time. As a result, "OpenTrace" must be modified to prevent the use of Firebase Analytics. Second, OpenTrace now requires users to provide a phone number to access the app, which is validated and stored via the Firebase Authentication service. The choice to request user phone numbers (or other identifiers) is likely motivated by a yearning for contact trackers that can be used to enable authorities to notify contacts of those confirmed positive with the virus. Other designs alert those contacts to a positive test result but allow them to take action. This could imply a clear trade-off between users' privacy and contact tracing efficacy. If it is determined that storing phone numbers is necessary, it is then also necessary to modify OpenTrace in such a way that Firebase Authentication is not used. Finally, OpenTrace's reversible encryption depends on a secret key held in a Google Cloud service, which makes it vulnerable to its exposure. Furthermore, centralized contact tracing methods can be divided into sub-categories based on the technology used or the contact tracing approach used. These sub-categories are the GPS-based technique as well as the Bluetooth-based privacy-enhancing contact tracing technique.

Centralized GPS-Based Contact Tracing Solutions

Timestamps of people's GPS-based locations can be used to construct their mobility at a specific point in time. Two people identified to have earlier contact can be re-analyzed whenever either test positive for COVID-19. Thus, the other individual can receive notices on the contact incident. Thus, the latter can be traced, inspected, and subjected to isolation. The method through which the authorities gather and exchange this data defines whether the system is centralized or decentralized. An example of GPS-based contact tracing solution that adopts the centralized model includes, the Chinese Alipay Health Code

contact tracing system, which assigns individuals color QR codes that determines if they are infected or healthy. A similar system is also the Corona 100 m (Co100) of South Korea. When users come within a hundred meters of an area recently visited by a COVID-19 patient, the app/software uses government-acquired location data to inform them.

Centralized Bluetooth-Based Contact Tracing Schemes

The Bluetooth-based contact tracing systems do not provide for collection of users' exact location information, as is the case with the GPS-based tracing apps. This, to a large extent, gives the users some sense of privacy and kills most of their anxieties. Because Bluetooth signals do not rebound and pass through most soft barriers; they offer superior contact tracing accuracy to GPS-powered apps. This helps to reduce false positives, especially when persons who are close to each other are viewed as a "contact" event when they are actually separated by walls. The Singaporean Bluetooth-based mobile phone app "TraceTogether" is an example of a centralized contact tracing system that uses the centralized paradigm. Regardless of their health situation, all app users must disclose their tokens as well as their phone info to the authorities. When an individual is diagnosed with the epidemic, they must notify authorities of their condition and disclose their contact information. The authorities then cross-reference every token in the contact token set with its token database, notifying users of matches via their phone contacts. For infected people, the privacy risk is similar to the centralized model in the GPS-powered system. Furthermore, because the authorities have each user's phone information as a unique identifier for connecting with other data repositories that may include delicate/private info about the users if they believe it is necessary, by and large, in the centralized model, users must find a way of trusting the government or designated authorities to keep their sensitive information protected.

In a nutshell, contact tracing data with a centralized model requires collection, integration, and sharing of data between targeted mobile nodes using some form of authorities. Thus, such models operate as a mass of surveillance systems that are connected and secured. The data are shared between each node with a security measure that each participant has a unique identifier and knows that target (when about) to share information. Users have no privacy when it comes to exchanging information with authorities/government officials, and they must trust the government to keep their information safe and secret.

3.7.2. Privacy in the De-Centralized Contract Tracing Model

Decentralized tracing models do not require aggregation and storage of individuals' information in some central repositories and are believed to provide a higher level of privacy preservation on users' information than the centralized approach. Thus, everyone is free to keep his/her information by themselves, on their own devices. More often than not, public platforms are developed to allow individuals the freedom to find out information such as the geographical spread of the epidemic in their locality. This information is passed through a diverse form of filtering, anonymization, and privacy tests before being shared publicly. Motivated by the fact that most existing contact tracing apps are typically privacy-intrusive, Hekmati et al. [35] proposed "CONTAIN", an early privacy-sensitive contact tracing approach. The app is a privacy-focused Bluetooth-based mobile digital contact tracing framework that does not rely on infrastructure-based location sensing or continuous storage of personally identifiable data. CONTAIN's purpose is to let users ascertain in total privacy whether or not they have been near someone who is sick. The work found and demonstrated CONTAIN's privacy promises. This study also included a simulation study that used an empirical trace dataset to show that by turning on the app in more crowded environments, users can increase their chances of detecting if they were near an infected person. Recently, a contact tracing tool adopting DP-3T and TCN protocols were built under a cooperation venture by Apple Inc. and Google Inc. The tool is meant to assist health agencies in reducing the pandemic spread emphasized user privacy and security through its design. The prominence of the two firms at the global level provided a

potential wider adoption of the anticipated technology, with detailed specifications by the firms. Nevertheless, the provision of relatively unknown/unspecified distributed scenery for contact tracing could enable typical assaults/attacks on the technology.

Bradford et al. [97] investigated a planned Apple/Google Bluetooth-based contact tracing interface for iPhone and Android smartphones, in terms of the exposure notification system's compliance with privacy and data protection laws. From their investigation, the GDPR's broad breadth was found to be advantageous in uncertain situations like a pandemic, rather than a disadvantage. A fundamental right-compliant systems design is supported by the principle-based approach that it supports. Narrower, sector-specific standards, such as the US Health Insurance Portability as well as Accountability Act (HIPAA) and even the new California Consumer Privacy Act (CCPA), on the other hand, leave loopholes that could be difficult to close in an emergency. Russo et al. [69] stated the possibility of solving the problem of data collection for the current pandemic with a privacy-preserving application that individuals can use on personal handheld devices such as smartphones while keeping their privacy. Furthermore, data acquired in such a fully distributed context aids in the containment of the pandemic, with particular care for users' privacy. As a result, they created the "WeTrace" contact tracing application. This method and application make use of the Bluetooth low-energy communication channel, which is available on many modern mobile devices, and employs public-key cryptography to allow messages to be deciphered for the intended recipient. Any arbitrary attack will even fail because every other possible participant simply listens to any input. Due to its inability to discern the real sender, "WeTrace" guarantees that any recipient of communication knows that it is for him or her.

Decentralized GPS-Based Contact Tracing Solutions

The MIT-led "Safe Paths" privacy-preserving platform is an excellent example of a decentralized contact tracing [134] architecture using GPS as the underlying technology. It consists of a smartphone software called PrivateKit as well as a web site named "Safe-Places." This software shares infected people's location records, which are anonymized and obfuscated, whereas PrivateKit enable people to compare their location records with the info given on SafePlaces. All this implies that uninfected persons do not have to share their location histories with authorities. Nevertheless, in the event that an individual is confirmed COVID-19 positive, his/her location history will have to be reported to the government. Now, since this location history might contain personal sensitive information, some form of filtering is done before the information is shared publicly. Similarly, the Israeli Hamagen app, which allows users to compare their GPS data alongside the country's central database of COVID-19 hotspots, is a suitable example.

Decentralized Bluetooth-Based Contact Tracing Solutions

There also exist Bluetooth-based contact tracing solutions that follow the decentralised contact tracing approach when it comes to the issue of users' data handling. The "Private Automated Contact Tracing" (PACT), "COVID Watch", and "COVIDsafe" apps, as well as the "Pan-European Privacy-Preserving Proximity-Tracing" (PEPP-PT) systems, are all noteworthy instances in this category. Over 400 volunteers from the United States, Australia, and Canada, in addition to other nations, contributed to the "COVID Watch" app. It uses private and local Bluetooth signals to provide anonymous COVID-19 exposure alerts/messages. PACT was created by MIT, whereas "COVIDsafe" is a contact tracing tool developed by the government of Australia. PEPP-PT is also a multi-component corporate program that sends alerts/messages regarding potential exposures. Anonymity or privacy protection can be strengthened by swapping produced random tokens across users to better thwart linkage to privacy violations. Apple and Google's privacy-protection cryptography solutions use secure multi-party computing to send anonymously encrypted messages without depending on a secure server [35,121].

### 3.7.3. Privacy in the Hybrid Contact Tracing Approach

A lot of debate surrounds the suitability of centralized and decentralized contact tracing strategies. Whereas some nations have adopted centralized contact tracing because they believe it is the most effective and simple method, others have chosen a decentralized approach, allowing people or users the majority of authority. Both of these strategies have their shortcomings, hence the need to strike a balance between centralized and decentralized. To this end, Ho et al. [38] presented pHyCT, a hybrid solution that provides fail-safety, privacy, and security. This system functions as a decentralized system, with users' identities being hidden from the central authority. In the event of infection, however, the affected user and the central authority can only expose the identities of users who have come into close contact. This feature allows it to deal with a situation where there are too many non-compliant users who fail to report after being exposed to an illness. Users who have not had any direct touch with an infected person are kept anonymous. Furthermore, [43] suggested a privacy-protecting contact tracing software. If a user has been close to a confirmed patient, the app allows them to be notified. According to the authors, their protocol is the most balanced and comprehensive privacy-preserving contact tracing method to date, striking a balance between security, privacy, and scalability. It allows all users to hide their past location as well as contact history with the government in terms of privacy. Nonetheless, all users can see if they were in close contact with a confirmed patient without discovering the patient's identity. To ensure that user privacy is respected, a zero-knowledge protocol was implemented. No user can submit a phony message to the system to initiate a false positive attack in terms of security. A formal security model and proof for the proposed protocol were provided. This can work well on resource-constrained devices.

Furthermore, a hybrid of centralized and decentralized models called pHyCT was proposed to guarantee fail-safe, privacy, and security of users' information. When a user is infected, it is only that infected user and the central authority that can expose the names of the users with whom they have had close contact. This feature allows it to deal with a situation where several non-compliant users fail to report after being exposed to an illness. Similarly, the "ROBust as well as privacy-presERving proximity Tracking" (ROBERT, France, Europe) technique is founded on federated server infrastructure and transient anonymous IDs, which can provide significant security and privacy assurances. Furthermore, the majority of existing systems can only trace human–human contact but not human–object contact (disease transmission through open surfaces). As a result, the "Private Contact Tracing" (PCT) system which is based on trajectory and reliable hardware were developed in Kyoto University Yoshida-Honmachi, Japan [40]. To develop a safe, efficient, and flexible PCT system, the development was founded on trusted Intel SGX hardware, and a new algorithm. Vaudenay [51] discovered that the basic decentralized proposal (DP-3T) contact tracing system is vulnerable to relay or even replay attacks. Thus, he presented more complex interactive techniques that avoid these attacks without sacrificing too much privacy. Interaction is unfortunately challenging for this application due to efficiency and security concerns. The countermeasures proposed thus far are either impractical or compromise crucial privacy elements [48]; therefore, a modest (improved) version of the model that is secure against common attacks such as relay as well as replay attacks was proposed that ensures privacy and upholds efficiency. To accomplish these goals, this study established the idea of "delayed authentication," which deals with a two-phase authentication code verification.

Although most smartphone-facilitated digital contact tracing protocols utilize the local "BLE" procedure for the identification of contagion-related propinquity, along with crypto protections, as required to improve the privacy of the system's users, they do not adequately shield unhealthy individuals from having their information leaked to their contacts, raising stigmatization fears. A study by Beskorovajnov et al. [25] offered a novel and realistic method with robust privacy assurances against active attackers to address privacy. It is built on the upload-what-you-saw paradigm, which incorporates a server-side separation

of roles and a technique to prevent users from determining which happenstance generated a notice with high-time resolution. This research also demonstrated the security of their proposed protocol using a simulation-based security concept of digital contact tracing in a real-world context. Mobile Health Data Collection Systems (MDCSs) are increasingly being used by community health workers in primary care programs to report their activities and perform health reviews, instead of paper-based systems. Because mHealth systems are characteristically privacy invading, it is critical to notify individuals and seek their consent to maintain their privacy rights. Another study by Iwaya et al. [37] presented an e-Consent tool designed specifically for MDCSs. It was built using the "Participant-Centered Consent toolkit" and the "Consent Receipt" specification, as well as a need assessment of consent management for data privacy. Usability research was conducted on the e-Consent solution. The findings of the study suggest that the design is effective at enlightening people about the kind of data processing as well as helping them make good decisions, and [54] also developed the PPMF, an e-government "Privacy-Protecting Mobile as well as Fog-Computing" platform that can track infected and suspected cases across the country. To track community transmission while retaining user data privacy, the researchers deployed individual/private portable devices with contact tracing apps as well as two kinds of fixed fog nodes, known as "Automatic Risk Checkers" (ARC) and "Suspected User Data Uploader Node" (SUDUN). When a user registers on the central application, their portable device obtains a "Unique Encrypted Reference Code" (UERC). The centralized app and the portable device both produce a "Rotational Unique Encrypted Reference Code" (RUERC), which would be transmitted via BLE. The ARCs are positioned at building access points to detect whether there are any positive or suspected cases nearby. If a confirmed case is discovered, the ARCs send out pre-warning signals to neighboring residents without exposing the afflicted person's info. The SUDUNs are installed in health centers and transmit test outcomes to a centralized cloud app. The data is then utilized to create a map that shows the difference between infected and suspected cases. As a result, governments can use the proposed PPMF framework to allow companies to continue their economic activity without being completely shut down.

Issues with privacy preservation of users are treated in relation to access control, time duration for which collected information can be stored or used by authorities, amount and type of data to be acquired from users, and who takes control of the data or application. In the different countries studied, any of the centralized, decentralized, or hybrid Contact Tracing Control Models were adopted. Countries such as India, Singapore, Australia, Canada, and the United Kingdom used the centralized model for their CT apps. This implies that the control over the contact tracing procedure lies with the central government authority. Adequate data can thus be collected for better policymaking and spread control. But users' privacy cannot be guaranteed beyond government promises. On the other hand, countries such as the United States, Saudi Arabia, and The Netherlands adopted the decentralized CT model, where users are given a large portion of control over their own data. In this case, guarantee over users' privacy is higher than in the centralized model. However, inadequacy of data may result. Now, most of the apps developed by the different countries fall under any of these three models. In addition, countries such as Australia, Germany, and even India carry out mass sensitization on what data the citizens are required to provide, and also make plain or clear, what other privacy preservation approaches (such as anonymization, cryptography, etc) they are implementing to safeguard users' data. For instance, In Australia's CT app "CovidSafe" functional encryption (Cryptography) was used alongside the Bluetooth Low Energy (BLE) technology. Even Germany and India adopted cryptography techniques as privacy measures. This is in order to fulfill the dictates of the GDPR Policy.

## 4. Results and Discussion

Altogether, 117 research papers were selected for this methodical review. This represents 93.60% of the articles that were considered for full-text review and that satisfied

the purpose (i.e., objectives) of this review study; hence, the remaining eight articles were excluded. In total, 86.32% (n = 101) of the articles were published in the year 2020, which indicates the immediate response that was given to studying the new pandemic upon declaration. As shown in Figure 2, these studies were carried out in a total of 41 countries. Some articles covered more than one key consideration simultaneously and by counting in this respect, a total number of 179 studies were realized. Overall, the African continent has a low number of studies (2.79%, n = 5) that pertains to development and deployment of contact tracing systems for COVID-19 prevention and containment all through the pandemic. More studies were reported from the United States (20.67%, n = 37) followed by studies done in China (9.50%, n = 17) and the United Kingdom (8.94%, n = 16) compared to what has been reported from the other countries. Similarly, it was observed that the United States gave approximately equal focus to the three different sensitive issues, i.e., the development considerations (32.43%, n = 12), contextual implications (32.43%, n = 12), and ethical issues (35.14%, n = 13) based on the articles investigated in this study. Apparently, a significant higher weight was given to ethical issues in the country. The situation is not remarkably different in China as the data reveal values of 35.29%, 35.29%, 29.41% for the three key considerations, respectively. Of the three main categories that were considered, it was found that more research efforts (47.86%, n = 56 articles) focused on design and implementation strategies to facilitate effective contact tracing systems. During the short pandemic time, a good number of automated and semi-automated tracing apps have been deployed especially in developed countries. Unlike using manual contact tracing apps during previous outbreaks like the Ebola virus, the recent development could be understood to have rapidly improved with the recent development and widespread of technology such as Bluetooth and location-based technologies. These can offer more possibilities such as preventing global viral outbreaks in the near future as well. In addition to having more study focused on design and implementation strategies, a good number of studies were also published on ethical and security issues. Some articles related to the subject matter that focused on ethical and security issues were published as opinions and commentary articles. Another trend found that researchers considered the related privacy, security, and ethical issues while developing and deploying contact tracing systems to be used in different countries. As reported in Figure 3 and Section 5, the 41 studies (35.04%) in this category also focused on development and/or evaluation of contact tracing apps, but with emphasis on users' data security and privacy. Meanwhile, a total of 15 (excluded) studies appeared as review or commentary articles.

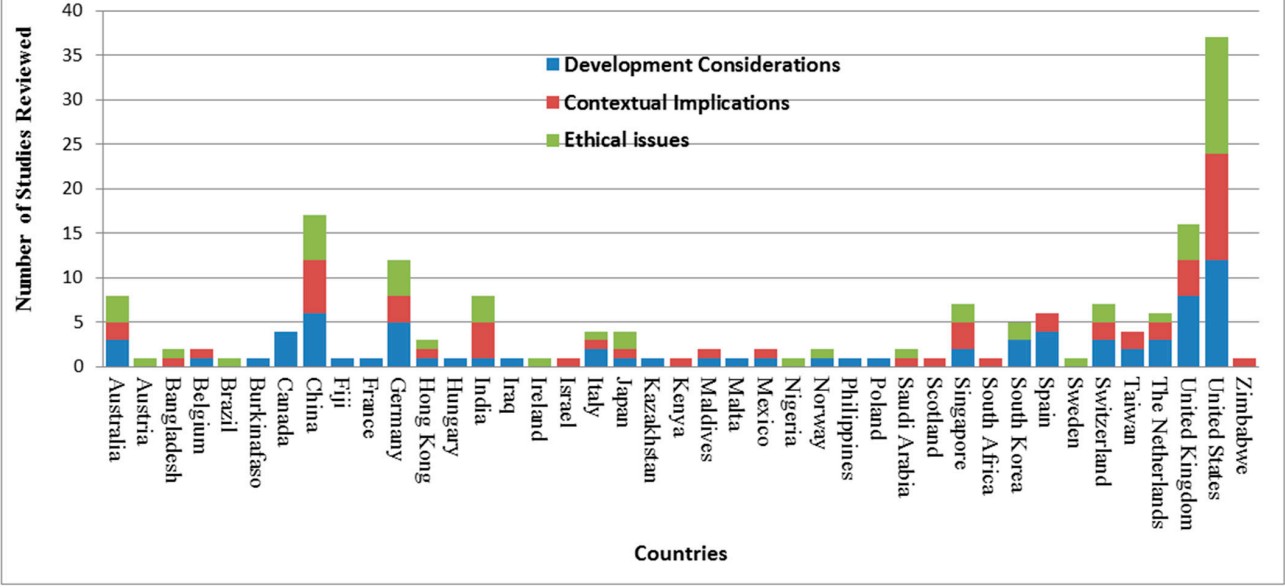

**Figure 2.** List of countries that contact tracing studies were reported for COVID-19.

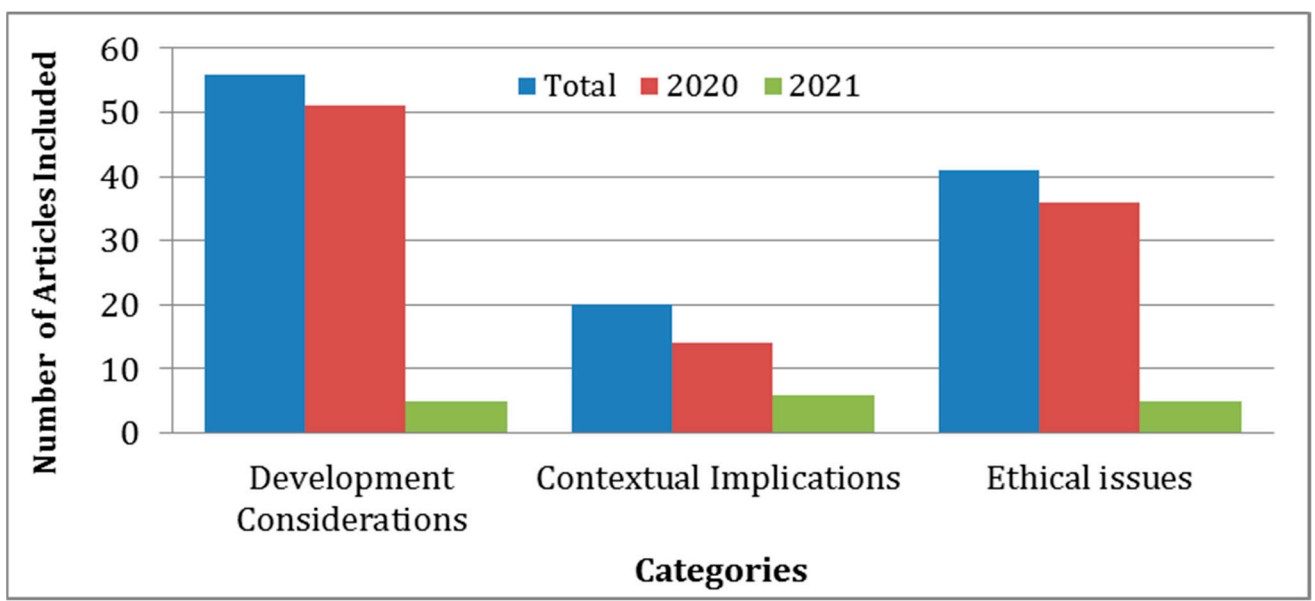

**Figure 3.** Categories of contact tracing studies.

Furthermore, it was observed that authors do have very diverse study motivations and views characterized in Figure 4. For instance, only 2.56% (n = 3) of the included studies focused purely on ethical issues. The ethically related studies (a total of 13 articles, 11.11%) were studies done with developmental perspectives. Hence, we realized that some studies are characterized with single and multiple sensitive issues as shown in the inner circle in Figure 4. For instance, whereas a larger part of the studies (n = 13) focused on design considerations solely, some other studies had a focus on perspectives of both design/cost considerations, and other's design/implementation consideration.

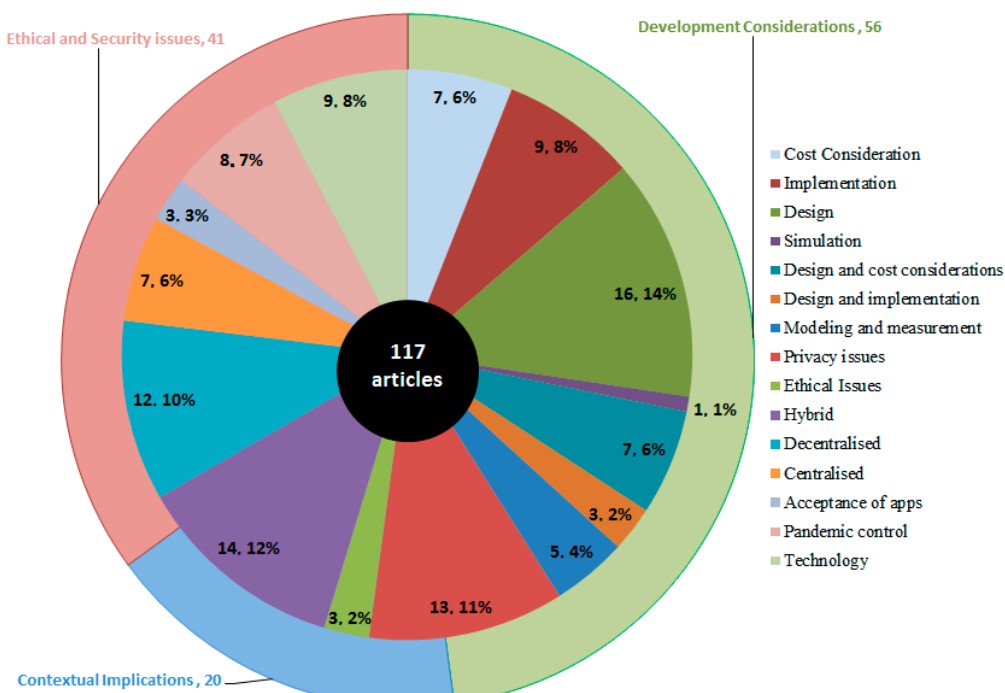

**Figure 4.** Analysis of the different issues focused on the included studies.

Tracing the contacts and locations of people identified as carrying the coronavirus is essential to preventing or reducing the further escalation of the pandemic. However, tracing

infected people should be done with the privacy and security of users' information to encourage participation and trust in the system to effectively combat the dreaded pandemic. In conclusion, existing literature shows that Asian countries prefer the centralized contact tracing strategy, whereas the US and European countries prefer the decentralized option. Contact tracing software or mobile apps have been developed and deployed in countries across the world, with varying degrees of concern relates to privacy. Whereas some contact tracing systems use GPS technology to collect data about users' whereabouts, Bluetooth-based systems generally rely on users' relative tempo-spatial proximity. As a result, less personal information from users is necessary or gathered. What is true is that the levels of anonymity and privacy-preservation techniques and features in both centralized and decentralized contact tracing models differ.

Our findings demonstrate that location tracking systems as well as communication technologies can be used to improve the delivery of COVID-19 infection automated contact tracing systems. These contact tracing technologies were classified by the World Health Organization [94] into three groups: outbreak response tools, proximity response tools, and symptoms response tools. As presented in Figure 3, 86.32% (n = 101) of the works included in this study were published in 2020. It could be said that more attention was directed to contact tracing systems in the year due to government investment in COVID-19 research, the priorities set by many publishers to encourage researchers, and the luxury of time available to researchers during the lockdowns imposed due to the pandemic. The COVID-19 pandemic, in general, impacted science in 2020 and changed research publishing. COVID-19 accounted for almost 4% of the world's research output last year. However, due to worldwide lockdown rules that compelled many researchers to stay at home, manuscripts on all subjects were submitted to scientific journals in record numbers in 2020. Only 13.86% (n = 14) of the articles disseminated in the year focused on contextual considerations and implications, whereas 50.50% (n = 51) and 35.64% (n = 36) involved studies on development and privacy issue considerations, respectively (details in Sections 3 and 5, respectively).

Overall, these articles cover 15.23% of the initial data retrieved during the database searches done for this study. Thus, this report has a wide horizontal coverage as, in essence, results of any arbitrary search on the repositories considered would produce at least 1 of the articles included in this study for every 6.67 articles returned from the database search. Outbreak response tools are applicable for inceptive localized outbreak management, cluster investigations at the initial stage, and limited populations' environment. These tools facilitate components of contact tracing procedures (such as case detection, data analysis, and management as well as listing and tracing of contacts), and they are specifically designed and valuable for public health response officers that are directly involved in contact tracing practices as well as outbreak research. They are usually designed with open access tools and open-source software that encouraged increased transparency, and continuous improvement of the systems. Finally, on privacy and ethical implications, 10.25% (n = 12) of the studies applied decentralized contact tracing solutions for location tracking and analysis whereas 5.98% (n = 7) applied centralized contact tracing solutions system development. A hybrid of both was applied in another 11.96% (n = 14) of the articles.

Proximity response tools, on the other hand, used GPS or Bluetooth signals to aid the identification of contacts by detecting when people were in close connection with an infected person for a long time. Individuals must have an active smartphone and keep it in their hands at all times to use proximity tracing techniques. People without smartphones may be left out of techniques that largely depend on proximity-tracing apps. As a result, proximity-tracing technologies must be supplemented with additional methods of contact tracing. Wearable GPS or Bluetooth devices may perhaps be developed for people who do not have cellphones or who want to use them more frequently. The revelation of location history, and possibly other private data raises numerous privacy concerns. With location-based techniques, privacy concerns and data protection must be carefully

examined. The third category includes symptom tracking tools, which are intended to collect self-reported signs and symptoms from users regularly to assess illness severity or the risk of infection with COVID-19. Data from self-reporting symptom tracking tools must be combined with data from other surveillance and monitoring systems. Because symptom monitoring systems are restricted in their ability to provide differential diagnoses, they must be utilized with care to avoid increasing the danger of negative medical outcomes for ailments not covered by the instrument. The efficiency of digital contact tracing technologies in assisting contact tracing is largely determined by the underlying technology's design and implementation method, as well as the level of adoption, confidence, and trust the society has given it. According to recent studies, to be optimally successful for contact identification, a digital contact tracing system should be used by 60% to 75% of a country's population [85,95]. Users' acceptability and adaptability, smartphone penetration in a community, and technological restrictions such as distance estimate error and defects in Bluetooth-based proximity detection are all obstacles to efficient usage of digital contact tracing for COVID-19 responses.

Findings from our study show that viable technological platforms are important for developing effective ACTs tools for COVID-19. Most contact tracing tools examined in this study rely on the use of existing mobile technology frameworks for deploying mobile tracking apps for COVID-19, and solely use Bluetooth and WIFI as communication channels. Developing wearable devices for automatic contact tracing for monitoring the spread of COVID-19 is an open research area that could be explored to overcome some of the challenges posed by using existing mobile technological platforms.

In addition, privacy and security concerns have been a major limitation to public acceptance of ACTs tools for COVID-19 tracking. Although several algorithms and models have been proposed in the literature to improve data security of IoT devices, dealing with privacy issues is still an open research area. Users are always curious about their information collected by IoT devices such as ACTs tool for COVID-19 that such personalized data will not be used without their consent. Developing more robust trust and confidentiality schemes for ACT tools could assuage privacy concerns.

Furthermore, we realized from our study that contextual matters such as cultural beliefs, population, and other social factors could affect users' acceptability of COVID-19 ACT tools. In some countries, governments have enforced it on the people by enacting new laws, whereas some have used some kind of incentives to encourage users to embrace such tools. Our findings show that including additional features such as personalized AI health analytics, GIS information services, and location-based social network information for users could encourage more acceptability of ACT tools for COVID-19 tracking.

## 5. Conclusions

This study addresses the sensitive issues that were focused on to prevent and contain the spread of COVID-19 upon being declared a pandemic. We conclude that contact tracing was globally found useful in controlling the spread of the virus. This scoping review identifies that the participation of different and multiple stakeholders helped in protecting privacy, security, and ethical concerns. Nevertheless, new technological methods were available for acquiring useful data and tracing contacts of confirmed COVID-19 victims. A high number of COVID-19 publications were found in 2020 and this can be related to the priorities set by the government of many countries and journals.

Results from our analysis indicated that the United States (20.67%, n = 37), United Kingdom (9.50%, n = 17), and China (8.94%, n = 16) had the highest number of studies reported, and development consideration, contextual implications, and ethical issues were the major concerns in developing the COVID-19 contact tracing application. Comparatively, the United States gave higher consideration to ethical issues (35.14%, n = 13), overall, and more attention was given to the design and implementation to facilitate effective contact tracing. Furthermore, the trend in research (35.04%, n = 41) focused on privacy, security, and ethical issues while developing and deploying contact tracing. The US and European

countries prefer a decentralized implementation approach, whereas the Asian countries prefer a centralized one. This review study's findings could lead to better measures for preventing worldwide epidemics. For COVID-19 prediction and containment, we envision the development of an effective automated contact tracing system with privacy-preserving and case-based referral characteristics, particularly in the African context. The study was based on a search period between November 2018 and May 2021. Thus, it did not cover the recent studies that were carried out on new variants of COVID-19. For speedy and accurate search results, the created search technique was confined to employing PubMed, Google Scholar, and ACM databases. This could have resulted in qualifying articles being excluded from other databases. Furthermore, due to the vast number of articles assessed, it may be possible that valid publications that satisfied the inclusion criteria were overlooked. This review did not include any non-English articles.

In the future, a more practical study will be carried out to evaluate the results obtained from the current analysis. We have studied the general requirements of automating contact tracing applications in terms of privacy issues, design considerations, and contextual implications in developed and developing settings. This is necessary to elucidate an overview of the implication of automated contact tracing to researchers in this domain. The results will be important in providing insight and knowledge-driven pursuits for government, medical professionals, researchers, and designers. Nevertheless, from the study, we have discovered that the situation with the different parameters varies for different countries; for instance, there is more concern for the security and privacy of users in the US and Europe compared to Asia, so for practical implementation in specific contexts, it will be essential to investigate the specific situation in each environment/ country. Therefore, we will be carrying out an analysis of user requirements for automated contact tracing applications in Nigeria as future work. Consequently, we intend to utilize the results from the analysis to develop a personalized automated contact tracing application that will integrate IoT technologies, a wearable sensor, and a multi-modal machine learning algorithm.

**Author Contributions:** Conceptualization, B.A.O.; methodology, B.A., O.A.S., O.O., U.M.C. and A.J.G.; data curation, O.A.S., A.J.G., O.O., A.E.T., T.I., A.A., O.B., T.A. and O.A.; writing—original draft preparation, O.A.S., A.J.G., O.O., A.E.T., T.I., A.A., O.B., T.A. and O.A.; writing—review and editing, B.A.O., B.A., T.Y., O.A.S., A.J.G. and O.O.; supervision, B.A. and B.A.O.; project administration, B.A.O. All authors have read and agreed to the published version of the manuscript.

**Funding:** The study is being supported through the Nigerian Tertiary Education Trust Fund, (NRF/SETI/ICT/00029).

**Data Availability Statement:** The data used for this study are available on reasonable request from the corresponding authors.

**Conflicts of Interest:** The authors declare no conflict of interest.

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
