# Peer review of "Contact Tracing Strategies for COVID-19 Prevention and Containment: A Scoping Review"

_2504-2289, doi:10.3390/bdcc6040111_

Round 1

Reviewer 1 Report

The authors searched and consulted a large number of literature, and reviewed related papers. But there are the following problems:

1. The overall organizational structure of the paper is unclear and needs to be reorganized.

2. The contribution of the paper is not enough. It is difficult for researchers to gain clear insights from papers.

In my opinion, this paper can be improved from the following aspects:
1. In the introduction, add a paragraph to summarize the contribution of the paper.

2. Add legends for each color in Figure 2. The correspondence between country names and histograms is not clear enough. Please make sure that the countries mentioned in the paper are shown in Figure 2.

3. In the conclusion section, the author should emphasize the research results of the paper and the prospect of future research focus.

Reviewer 2 Report

Authors proposed a review on the key elements of digital contact tracing strategies for COVID-19 prevention and containment. Search strategies are clear, and these have been reported in Section 2.1.

The introduction is exhaustive; however, I suggest extending the references of the statements related to the limitations applied by governments, and to the model for trend forecasting and estimation. To give an example, I suggest mentioning PMID:35885152, it refers to epidemiology of Covid-19 in several countries (USA, Italy, France, Sweden, UK), as well as to the trend comparison based on limitations (e.g., lockdown). 

Section 2 (Materials and Methods) is introduced with the strategies used to investigate the topic on the public literature. It also reports the ethical considerations. The section is written based on the following main topic: implementation, efficiency, contextual implications, privacy and security. It is very exhaustive, even if fragmented.

The paper focused on a sub-set of countries (e.g., Australia, Brazil, Malta, Sweden, Taiwan); authors should explain how these are chosen. Other countries have been at the center of debates related to contact-tracing, not frowned upon, e.g., UK, Spain, USA, France, …

To give an example, authors reported “More studies were reported from the United States (20.67%, n = 37) followed 1030 by studies done in the United Kingdom (9.50%, n = 17)..” (Section 3, lines 1030-1031), but UK is not shown in Figure 2.

Authors could clarify the situation for these countries, explaining their ethical considerations related to privacy, mostly.

Section 3 reports the data related to the mentioned papers. It reports only information related to the number of papers for the topic. In my opinion, the results should be in accordance with methods, e.g., by reporting information also on the implementation, efficiency, privacy and security. Some information is missing.

In my opinion, I think that this manuscript is very interesting, however, the results (and discussion) should be revised to propose a study related to the topic reported within methods.

Minors: The manuscript contains typos and grammar mistakes.

Round 2

Reviewer 1 Report

Dear authors,

I have pointed out some problems in the paper and given some suggestions in the previous review. I'm glad to see that the paper has been greatly improved in this version.

1. The paper's overall organizational struction is clear and understandable.
2. The contribution of the paper has been added.
3. The authors have updated the conclusion section to emphasize the research results and the prospect of future research focus.

I have no further comments and I would like to accept this work.

Reviewer 2 Report

Authors proposed a revised version of the original manuscript. I comment this one as follows.

The introduction is exhaustive.

Section 3 (Materials and Methods) describe the strategies used to investigation, by focusing on the methodology and its application. The section is written based on the following main topic: implementation, efficiency, contextual implications, privacy and security.

Furthermore, the authors extended the manuscript introducing a more-detailed explanation on the privacy related issues, and other relevant information.

In the revised version, both Result and Discussion have been revised; these ones support the proposed methodology.

Authors responded to all my comments. In my opinion, the manuscript is very interesting, and I suggest it for acceptance.